# Role of Two Plant Growth-Promoting Bacteria in Remediating Cadmium-Contaminated Soil Combined with *Miscanthus floridulus* (Lab.)

**DOI:** 10.3390/plants10050912

**Published:** 2021-05-02

**Authors:** Shuming Liu, Hongmei Liu, Rui Chen, Yong Ma, Bo Yang, Zhiyong Chen, Yunshan Liang, Jun Fang, Yunhua Xiao

**Affiliations:** 1College of Bioscience and Biotechnology, College of Resource and Environment, Hunan Agricultural University, Changsha 410128, China; liushuming@stu.hunau.edu.cn (S.L.); hongmeiliu@stu.hunau.edu.cn (H.L.); huangyuanhao321@163.com (R.C.); mayong@stu.hunau.edu.cn (Y.M.); byang@hunau.edu.cn (B.Y.); zhiyongchen@hunau.edu.cn (Z.C.); lyss3399@126.com (Y.L.); 2Hunan Engineering Laboratory for Pollution Control and Waste Utilization in Swine Production, Changsha 410128, China; 3Key Laboratory for Rural Ecosystem Health in Dongting Lake Area of Hunan Province, Changsha 410128, China

**Keywords:** phytoremediation, metal(loid)s, bioinoculation, *Miscanthus floridulus* (Lab.)

## Abstract

*Miscanthus* spp. are energy plants and excellent candidates for phytoremediation approaches of metal(loid)s-contaminated soils, especially when combined with plant growth-promoting bacteria. Forty-one bacterial strains were isolated from the rhizosphere soils and roots tissue of five dominant plants (*Artemisia argyi* Levl., *Gladiolus gandavensis* Vaniot Houtt, *Boehmeria nivea* L., *Veronica didyma* Tenore, and *Miscanthus floridulus* Lab.) colonizing a cadmium (Cd)-contaminated mining area (Huayuan, Hunan, China). We subsequently tested their plant growth-promoting (PGP) traits (e.g., production of indole-3-acetic acid, siderophore, and 1-aminocyclopropane-1-carboxylate deaminase) and Cd tolerance. Among bacteria, two strains, *Klebsiella michiganensis* TS8 and *Lelliottia jeotgali* MR2, presented higher Cd tolerance and showed the best results regarding in vitro growth-promoting traits. In the subsequent pot experiments using soil spiked with 10 mg Cd·kg^−1^, we investigated the effects of TS8 and MR2 strains on soil Cd phytoremediation when combined with *M. floridulus* (Lab.). After sixty days of planting *M. floridulus* (Lab.), we found that TS8 increased plant height by 39.9%, dry weight of leaves by 99.1%, and the total Cd in the rhizosphere soil was reduced by 49.2%. Although MR2 had no significant effects on the efficiency of phytoremediation, it significantly enhanced the Cd translocation from the root to the aboveground tissues (translocation factor > 1). The combination of *K. michiganensis* TS8 and *M. floridulus* (Lab.) may be an effective method to remediate Cd-contaminated soils, while the inoculation of *L. jeotgali* MR2 may be used to enhance the phytoextraction potential of *M. floridulus*.

## 1. Introduction

Metal(loid)s pollution is a serious problem that results from rapid industrial development, leading to the contamination of several ecosystems, including rivers and soils, further affecting human and animal health through the food chain. Among these toxic trace elements, cadmium (Cd) has attracted widespread concern due to its high toxicity, causing cancer incidence, osteoporosis, and even death [1]. China is one of the largest rice producers globally, and Hunan Province is one of China’s main rice-producing areas. Previous studies reported that many arable lands in Hunan Province are contaminated with Cd, and its content exceeded the national environmental quality standards [2,3]. Therefore, the remediation of Cd-contaminated soil in Hunan Province is an urgent task.

Many approaches have been applied to remediate Cd-contaminated soils, including physical, chemical, and biological techniques. Phytoremediation may include phytostabilization [4], phytoextraction [5], phytovolatilization [6], rhizofiltration [7], phytodegradation [8], and phytotransformation [9], which result in the removal and/or stabilization of metal(loid)s in contaminated soils. This approach has several advantages, such as being environment-friendly, causing no secondary pollution, and improving landscaping of degraded areas [10]. However, this method is limited by plant biomass, growth cycle, and metal(loid) availability [11]. Recently, some studies reported that chemical modifiers could increase the bioavailability of metal(loid)s to enhance phytoremediation [12]. In addition, mycorrhizal fungi can help plants to absorb nutrients and to improve the effectiveness of metal(loid)s’ uptake [13]. In addition, some plant growth-promoting bacteria (PGPB) could promote plant growth, change the availability of metal(loid)s, and promote its accumulation [14,15]. PGPB may promote plant growth through the production of indole-3-acetic acid (IAA), siderophore, and 1-aminocyclopropane-1-carboxylate (ACC) deaminase activity, nitrogen (N_2_) fixation, and phosphorus (P) and potassium (K) solubilization [15,16]. Additionally, PGPB have been known to increase the bioavailability of metal(loid)s [17]. Therefore, it is a promising approach to combine plants with PGPB to remediate soil metal(loid)s contamination.

*Miscanthus* spp. are energy crops (C4 plant) with high yield and carbon storage that can be used to produce renewable fuels [18,19]. It was reported that *Miscanthus* spp. could use N, P, K, Ca, Mg, and Na from municipal sewage sludge to obtain high biomass and remediate sewage sludge [20]. Besides, *Miscanthus* spp. are dominant plants in mining areas, such as Rongxi Manganese Mine in Chongqing, China [21], and Kabjeong coal mine in Korea [22]. Therefore, they have the advantages of being tolerant to several metal(loid)s and strong environmental adaptability. Previous studies showed that *M. sacchariflorus* (Maxim.), *M. sinensis* Andersson, and *M. floridulus* (Lab.) had the potential to remediate metal(loid)s contamination in soils, and the remediation efficiency was related to plants’ genotype and metal(loid)s type [23,24,25,26,27]. The ability of *M. floridulus* to accumulate Cd was significantly higher than that of *M. sacchariflorus* (Maxim.), as demonstrated by Guo et al. [23]. Phytostabilization efficiency, protein and chlorophyll content, as well as biomass of *M. sinensis* were improved by inoculating the PGPB *Pseudomonas koreensis* AGB-1 [25].

Recently, many researchers have focused on remediating soils contaminated with metal(loid)s using *Miscanthus* spp. However, to the best of our knowledge, interactions between *M. floridulus* (Lab.) and rhizosphere bacteria/endophytes have not been reported, and the effects of endophytes and rhizosphere bacteria on *M. floridulus* (Lab.) growth and metal(loid) uptake in Cd-contaminated soil are still unclear. Therefore, the primary purposes of this study were: (1) to isolate and screen rhizospheric bacteria and endophytes with plant growth-promoting (PGP) traits from dominant plants in Cd-contaminated mining area, (2) to explore the effects of the selected PGPB on *M. floridulus* (Lab.) growth, and (3) to evaluate the effects of selected PGPB on the Cd remediation efficiency. This research will help to further understand the feasibility of the PGPB–plants combination remediation approach in Cd-contaminated soils.

## 2. Materials and Methods

### 2.1. Sample Collection and Bacterial Isolation

Bacteria were isolated from the roots and rhizosphere soils of five dominant plants, collected from a Pb–Zn mining area with Cd contamination (Huayuan, Hunan, China). The center of the Huayuan Pb–Zn mining areas is at 109°21′ E longitude and 28°30′ N latitude. The dominant plants were *Artemisia argyi* Levl., *Gladiolus gandavensis* Vaniot Houtt, *Boehmeria nivea* L., *Veronica didyma* Tenore, and *M. floridulus* (Lab.). To isolate rhizosphere soil bacteria, approximately 1.0 g of rhizospheric soil was dissolved in 10 mL sterile distilled water (10^−1^), 0.5 mL 10^−1^ soil solution was added into 4.5 mL sterile distilled water (10^−2^), and the soil solution was diluted to 10^−8^ by ten times gradient successively; afterward, 100 μL diluted solution (10^−6^, 10^−7^ and 10^−8^) was spread on Luria–Bertani (LB) agar media, and incubated at 30 °C for 24 h. Bacteria were isolated and purified according to different colony morphologies and preserved in 20% glycerin at −80 °C. For endophyte isolation, plant roots were first surface-sterilized with 2.5% sodium hypochlorite for 10 min and then washed with sterile distilled water three times [28]. Sterile water (1 mL) of the third wash was cultured in 100 mL LB media for 8 h. If there was no bacterial growth, disinfection was completed. Subsequently, plant roots were homogenated and suspended in LB agar medium at 30 °C for 24 h. The steps of purifying and preserving the bacteria were the same used for rhizosphere soil bacteria isolation.

Based on the different colors, shapes, and sizes of bacterial colonies, 23 rhizobacterial strains (MS1–MS3, ZS1–ZS4, AS1–AS3, PS1–PS5, and TS1–TS8) and 18 root endophytic bacterial strains (MR1–MR4, ZR1–ZR3, AR1–AR3, PR1–PR4, and TR1–TR4) were isolated. “S,” soil, “R,” root, and the number represents the order.

### 2.2. PGP Traits Tests

The strains were screened for PGP traits, including IAA production [29], ACC deaminase activity [30], siderophore production [31], N_2_ fixation [32], P solubilization [33], and K solubilization [34]. IAA production was tested using Sackowski’s reagent (100 mL 35% HClO_4_ and 2 mL 0.5 mol/L FeCl_3_). For IAA determination, bacterial strains were grown in LB liquid medium containing 100 mg/L tryptophan at 180 rpm and 30 °C for 24 h, and then centrifuged at 10,000 rpm for 10 min. Finally, mixing 1 mL of supernatant with 2 mL of Sackowski’s reagent (*v*/*v*, 1:2) and reacting for 30 min in the dark, the optical density of IAA was measured at 530 nm [35].

Then, the bacteria, which could produce IAA, were screened for further ACC deaminase activity. They were cultured in a DF medium containing 3 mM ACC as the sole N source, 4 g/L KH_2_PO_4_, 6 g/L Na_2_HPO_4_, 0.2 g/L MgSO_4_, 2 g/L glucose, 2 g/L sodium gluconate, and 2 g/L citric acid [36]. After 24 h, the cultured bacteria were centrifuged for collection and washed twice using 20 mL of 0.1 mol/LTris-HCl (pH = 7.6), at 10,000 rpm and 4 °C. Then, to obtain the crude enzyme to determine protein concentration and α-ketobutyric acid (α-Kb) content, bacteria were resuspended in 600 μL of 0.1 mol/L Tris-HCl (pH = 8.5), adding 30 μL of toluene, followed by 30 s ultrasonication. Protein concentration was determined using Bradford’s method [37], and enzyme activity was determined using Hassan’s method [38]. The amount of 1 μmol α-ketobutyric acid formed per minute was defined as one enzyme activity unit.

Subsequently, the bacteria, which had ACC deaminase activity, were screened for the ability to produce siderophores. Siderophore production was qualitatively and quantitatively determined using the chrome azurol S (CAS) indicator solution. Strains were inoculated on CAS solid medium for 24 h at 30 °C. The color of the colonies in the medium from blue to orange indicated that siderophores were produced. The quantitative experiment of siderophore production refers to the method of Ghavami et al. [39]. The selected bacteria incubated in LB at 30 °C for 24 h were centrifuged at 10,000 rpm at 4 °C for 10 min. The 3 mL supernatant and 3 mL CAS indicator solution (*v*/*v*, 1:1) were mixed, and absorbance (A) was measured at 630 nm after 1 h. Uninoculated LB was mixed with CAS indicator solution (*v*/*v*, 1:1) as the control, and absorbance (Ar) was measured: siderophore unit (SU) = [(Ar − A)/Ar] × 100.

Finally, the bacteria, which could produce siderophore, were screened for N_2_ fixation [40], P solubilization [41], and K solubilization [34] on Ashby’s, Pikovaskaia’s (PKO; Ca_3_(PO_4_)_2_), and Aleksandrov (potassium feldspar) media with an Oxford cup, respectively. Then, 10 μL of bacterial suspension were inoculated into the Oxford cup of three different media at 30 °C for 24 h; then, the sizes of the colonies and the halos were measured. The N_2_ fixation, P solubilization, and K solubilization were expressed by the ratio of the halo size (D) divided by colony size (d).

### 2.3. Decreasing pH and Cd Tolerance of PGPB

The capacity for producing acid compounds was determined by measuring the pH value. Approximately 1 mL of bacterial suspension in LB liquid medium was inoculated into a medium containing 10 g/L of tryptone, 5 g/L of NaCl, and 10 g/L of glucose. The pH of the bacterial suspension was measured by pH meter (Bante210, BANTE instruments Shanghai Co., Ltd., Shanghai, China) every 24 h for 5 days. Each treatment was repeated three times. The maximal tolerance concentration (MTC) was determined by the highest Cd concentration where bacterial strains could grow (0, 0.09, 0.18, 0.36, 0.53, 0.71, 1.07, 1.33, 1.78, and 2.22 mM) at 30 °C for 24 h [42,43,44]. The Cd salt used to make the concentration gradient was Cd(NO_3_)_2_·4H_2_O. Each treatment was replicated three times.

### 2.4. PGPB Identification

Based on the above experiments, two strains (one from soil and one from roots) with strong PGP traits and a certain ability to produce acid compounds were screened for further identification. Firstly, we tested the growth of the selected strains under different Cd concentrations (0, 0.18, 0.36, and 0.53 mM) using LB liquid medium. Samples were obtained every six hours. The cell concentration was represented by the absorbance, which was measured using an ultraviolet-visible spectrophotometer (UV-1000 AOELAB, Aoyi instruments Shanghai Co., Ltd., Shanghai, China) at 600 nm. Three replicates were determined. Secondly, colony morphology was observed by incubating in LB solid plates and cell morphology was observed by Optical microscope through dyeing with crystal violet.

Phylogenetic identification was performed by using 16S rRNA. DNA extraction was performed following the standard procedure of the TIANamp Bacteria DNA Kit (DP302-02) (TIANGEN BIOTECH Beijing Co. Ltd., Beijing, China). The 16S rRNA region was amplified by PCR using primers 27F (5-AGAGTTTGATCACTGGCTCAG-3) and 1492R (5-CGGCTTACCTTGTTACGACTT-3). The PCR program was as follows: initial denaturation at 95 °C for 5 min, 30 cycles of 95 °C for 45 s, 55 °C for 45 s, and 72 °C for 1 min, and a final extension at 72 °C for 10 min. The sequences were determined by Sangon Biotech Co., Ltd. (Shanghai, China). The alignment was done with the BLAST program (NCBI database) in http://www.ncbi.nlm.nih.gov/BLAST (accessed on 11 January 2021), and a Phylogenetic tree was constructed using MEGA 5.2 [28].

### 2.5. Pot Experiments

Soil used in pot experiments was collected from Hunan Agricultural University, Changsha, China (unpolluted soil). A small amount of soil was air-dried and passed through a 0.15 mm sieve for analysis. Soil physicochemical properties such as pH, concentration of total N, and organic matter were measured according to the methods described by N’Dayegamiye et al. [45]. The content of available P and available K was performed following the standard procedure of the Acid soil available phosphorus Assay box (ZC-S0558) and Soil available potassium Assay box (ZC-S0567) (ZCIBIO Technology Co., Ltd., Shanghai, China).

The effects of TS8 and MR2 inoculation on growth and Cd uptake in *M. floridulus* (Lab.) plants were investigated in pots (15 × 9 × 11.5 cm) containing 2 kg of 2 mm sieved soil. Cd(NO_3_)_2_·4H_2_O was added to the unpolluted soil, forming polluted soil containing total Cd of 10 mg/kg. The Cd-contaminated soil was irrigated with 100 mL of water every two days and incubated for 21 days. There was a disk to prevent water losses and to protect the soil from metal(loid)s leaching. Plants grown for 30 days to 30–40 cm of height in uncontaminated soil were transplanted into artificially Cd-contaminated soil. The experimental design was as follows: (1) 10 mg Cd/kg soil without PGPB inoculation (CK), (2) 10 mg Cd/kg soil inoculated with TS8 bacterial solution, and (3) 10 mg Cd/kg soil inoculated with MR2 bacterial solution. Bacteria TS8 and MR2 were suspended in sterile ultrapure water, and the concentration of the bacterial suspension was 1 × 10^11^ cfu·mL^−1^. For inoculation, 10 mL of TS8 and MR2 bacterial suspension were added to treatment (2) and treatment (3), respectively. Three plants were transplanted in each pot, and each treatment was replicated three times. *M. floridulus* (Lab.) was provided by the Hunan Engineering Lab Ecological Applicational Miscanthus Resource. Each pot was watered with 100 mL every two days and placed in the outdoors for 60 days on 1 August 2019. For the climatic conditions during the planting period, the temperature was 25–40 °C during the day, 15–30 °C at night, and the humidity was about 50%.

### 2.6. Cd Uptake of M. floridulus

Plants were harvested after 60 days and washed with tap water, dipped in 20 mM of Na_2_EDTA to remove non-specifically bound surface Cd, and washed in distilled water three times. The height of *M. floridulus*, and dry weight of the roots, leaves, and stems were measured. Before measuring dry weight, plants were oven-dried at 105 °C for 30 min [43], and then dried at 80 °C for 24 h [25]. For determining Cd content, plant leaf, stem, and root samples were ground. Each sample (0.1 g) was digested with 8 mL of HNO_3_ (15.32 mol/L) and 2 mL of HClO_4_ (12.61 mol/L) (*v*/*v*, 4:1), covering the curved neck funnel, heating at 95 °C for 15 min, then at 120 °C for 20 min, then at 70 °C for 2 h, and finally at 190 °C until the remaining liquid was approximately 1 mL. Soil samples (0.5 g) were digested with 6 mL HCl (12.28 mol/L) and 2 mL HNO_3_ (*v*/*v*, 3:1), covering the curved neck funnel, heating at 80 °C for 1 h, and then to 150 °C for 1 h. After the digestion tube had been cooled, 2 mL of HClO_4_ was added and then heated to 190 °C for 2 h. Finally, the temperature was increased to 220 °C, and the solution was digested to nearly dry. After digestion, the volume was adjusted to 50 mL, and Cd concentration was measured using the graphite furnace atomic absorption spectrophotometer method (AAS-GF, AA-6880FG, Shimadzu (China) Co., Ltd., Beijing, China) [46]. For assuring and controlling quality, Cd content in the reference material GBW(E)083179 certified by the National Tobacco Quality Supervision and Inspection Center in China was determined (Appendix A). Then, we calculated bioconcentration factor (BCF) and translocation factor (TF) with the following formulas: BCF = the Cd concentration in plants/the Cd concentration in soil × 100; TF = the Cd concentration in aboveground parts of plants/the Cd concentration in plant roots × 100. BCF represented the Cd-accumulating ability of *Miscanthus* spp., and TF represented the Cd-translocation ability from the roots to the aboveground.

### 2.7. Statistical Analysis

All data provided in this study are the average values of the three replicates. Using SPSS 25, differences in parameters based on Tukey’s test were conducted by one-way analysis of variance (ANOVA) to determine significant differences at the 5% level.

## 3. Results

### 3.1. PGP Traits of Bacteria

The result of the IAA production test revealed that 16 strains had this capacity (Figure 1a). The highest IAA production was observed for strains ZS3 (74.21 mg/L) and TS8 (65.78 mg/L), whereas the remaining bacterial isolates produced less than 20 mg/L of IAA.

Subsequently, the ACC deaminase activities of these 16 IAA-producing bacteria were tested. We found that the TS8 strain (1.41 μmol α-Kb/(h·mg)) showed significantly higher ACC-deaminase activity production than the other 15 strains (Figure 1b).

Screening for siderophore-producing traits was based on strains that produce IAA and have ACC deaminase activity. Siderophore results indicated that ten strains could produce siderophores (MR2, MR3, MS3, PR4, PS3, TS8, ZR1, ZS1, ZS3, and ZS4; Figure 1c). The siderophore yields of ZR1 (39.48%) and ZS4 (39.47%) were significantly higher than those of TS8 (32.97%), PS3 (27.37%), ZS3 (28.22%), MS3 (22.93%), MR3 (18.36%), PR4 (14.97%), MR2 (14.03%), and ZS1 (15.14%).

The above ten strains were tested for their abilities of N_2_ fixation, P solubilization, and K solubilization (Table 1). Five strains (ZS1, ZS4, MR2, PR4, and ZR1) could solubilize P and K and fix nitrogen. One strain (TS8) could not fix nitrogen. Four strains (ZS3, MS3, MR3, and PS3) could not solubilize K. The strains PS3 (2.66) and MR3 (2.08) showed relatively strong capacities for N_2_ fixation. MS3 (3.39) and PR4 (3.35) had significant advantages in P solubilization. TS8 (2.33) and MR2 (2.09) showed a relatively strong capacity for K solubilization.

### 3.2. Cd Tolerance and pH Variation

The MTC of ten functional bacteria to Cd is shown in Table 2 and Appendix A. The results showed that ZR1 possessed the highest tolerance at 200 mg/L, and the MTC of MS3, MR2, ZS3, and ZS4 was 120 mg/L, while the other five strains exhibited the lowest MTC (80 mg/L).

Subsequently, pH values were tested to evaluate the ability of isolates to produce acid compounds. It was found that six strains, namely MR2 (4.15), MR3 (3.73), MS3 (5.40), PR4 (5.61), PS3 (4.24), and TS8 (5.61), could efficiently decrease pH values in the cultures on day 5 (Figure 2).

### 3.3. Characteristics and Identification of Selected PGPB

#### 3.3.1. Morphological Identifications and Phylogenetic Analysis

The morphological characteristics of the two selected strains were observed by simple staining. Results (Appendix A) indicated that the bacterial colonies of TS8 had a matt light-yellow surface, while the bacterial colony size of MR2 had a glossy white surface. The bacterial colony of MR2 was long-rod, whereas TS8 was short-rod (Appendix A). Selected PGPB strains TS8 and MR2 were identified by partial 1400 bp sequencing of the 16S rRNA gene. TS8 exhibited 99.52% homology with *Klebsiella michiganensis* Biosolid27, whereas MR2 showed 99.59% homology with the *L. jeotgali* strain PFL01 (Figure 3). The sequences of *L. jeotgali* MR2 (accession No. SUB8850737) and *K. michiganensis* TS8 (accession No. SUB8850709) were deposited in GenBank. To the best of our knowledge, this is the first time that *L. jeotgali* was isolated from *M. floridulus* roots, and PGP traits were determined.

#### 3.3.2. Effects of Different Cd Concentrations on PGPB Growth

According to the Cd-tolerance experiment, Cd background concentration was set to 0, 0.18, 036, and 0.53 mM to observe growth and tolerance of the strains. Figure 4 showed that TS8 was more sensitive to Cd stress than strain MR2. At the 36th hour, MR2 was inhibited under >40 mg/L Cd stress, while TS8 was significantly inhibited under 20 mg/L Cd stress. The growth of MR2 was reduced by 8.37%, 36.54%, and 63.88% at 0.18, 0.36, and 0.53 mM of Cd respectively, while the growth of strain TS8 was reduced by 64.53%, 63.44%, and 82.05% at 0.18, 0.36, and 0.53 mM of Cd, respectively.

### 3.4. M. floridulus Growth

To evaluate the effects of the two selected PGPB (TS8 and MR2) on *M. floridulus* growth, some parameters (plant height and dry weight of tissues) of the TS8-inoculated, MR2 inoculated, and non-inoculated plants under Cd stress were compared (Figure 5). The measures were determined in 60-day-old plants. The data showed increases (*p* < 0.05) of about 39.88% and 99.06% in plant height and dry weight of leaves of the TS8-inoculated plants (101.83 ± 7.68; 2.11 ± 0.45) respectively, if compared to the control. However, the TS8- and MR2-inoculated plants showed decreases (*p* < 0.05) in dry weight of stems of about 64.58% and 80.20% (0.34 ± 0.01; 0.19 ± 0.04) respectively, compared to the control (0.96 ± 0.06).

### 3.5. Cd Content in M. floridulus and Soil

The initial soil physicochemical properties were as follows: the pH value was 6.4, the available P concentration was 9.91 mg/kg, the organic matter content was 5.92 g/kg, the total N concentration was 0.32 g/kg, and the available K concentration was 10.79 mg/kg.

The pot experiments showed some effects of TS8 and MR2 inoculation on Cd accumulation in the roots, stems, and leaves of *M. floridulus* (Figure 6a). Results revealed that the Cd concentration in the leaves of the MR2-inoculated plants was 80.30% higher than that in TS8-inoculated plants. However, compared with CK, bacterial inoculation with both strains showed no significant influence on Cd accumulation in plant tissues.

The concentration of total Cd in soil is an essential parameter for determining the efficiency of phytoremediation. According to the results shown in Figure 6b, the soil Cd concentration in the TS8-inoculated group was 5.08 mg/kg, which was significantly (*p* < 0.05) lower than in the CK (5.45 mg/kg) and MR2-inoculated (5.77 mg/kg) groups. TS8 inoculation showed a significant increase in remediation efficiency of about 8.1% (49.23%) relative to the control (45.53%). The pH value showed no significant difference among three groups (Figure 6c).

BCF > 1 showed the plants’ ability to transport Cd from soil to plant, while TF > 1 represented a high ability of plants to transport Cd from the roots to the aboveground tissues [27,46]. The results (Table 3) showed that the TF in the MR2-inoculated group (1.25 ± 0.03) was significantly (*p* < 0.05) higher than that in the CK (0.82 ± 0.33) and TS8-inoculated (0.54 ± 0.20) groups; however, there was no significant difference in the BCF for the three treatments.

## 4. Discussions

Recent studies indicate that PGPB are an excellent choice to enhance phytoremediation efficiency of metal(loid)s [47]. In our study, two PGPB (MR2 and TS8) were isolated, screened, and inoculated in *M. floridulus* (Lab.) plants grown under Cd contaimination. TS8 had better PGP traits than the other strains (Figure 1 and Table 1). Additional studies demonstrated that endophytes with PGP traits might help enhance metal(loid)s phytoremediation. Endophyte Pseudomonas putida sp. RE02 promoted the root growth of Trifolium repens L. by 25.9% in Cd-contaminated soil [48]. MR2 was isolated from the roots of *M. floridulus* (Lab.), showing all PGP traits and a relatively strong ability to decrease the pH in vitro. Our results showed that TS8 could promote *M. floridulus* (Lab.) growth and Cd remediation, as showed by the significant increases in plant height and dry weight of leaves of the TS8-inoculated plants.

PGPB can secrete plant-growth hormones (e.g., IAA and ethylene) and provide nutrients (e.g., N, P, K, and Fe) [25] to improve plant growth. IAA and ethylene are essential hormones for plant growth: IAA can promote plant growth, while excess ethylene inhibits it. ACC is the substrate of ethylene synthesis, and ACC deaminase can hydrolyze ACC, reduce the synthesis of ethylene, and alleviate the stress from ethylene [49]. In the study by Babu et al., Pseudomonas koreensis AGB-1 produced 8.2 mg/L IAA [25]. Another study indicated that Enterobacter sp. strain EG16 could produce 3.14 μM of IAA [46]. Compared with previous studies, TS8 (65.78 mg/L) had stronger IAA production ability. Low iron content in plants hinders the development of chloroplasts and the synthesis of chlorophyll [50]. TS8 showed the capacity to produce siderophore, which was reported to be easily absorbed by plants and could be used as an iron source to promote plant growth [50]. Besides, TS8 had the ability to solubilize P and K, and may help to provide the essential nutrients (P and K) for plant growth. According to previous reports, Klebsiella michiganensis MCC3089 has a positive effect on increasing rice germination rate (EC50) under Cd stress [51]. Klebsiella pneumoniae MCC 3091 can also promote the growth of rice seedlings [52]. Burkholderia SaMR10 and Sphingomonas SaMR12 promoted the dry weight of Brassica. juncea by 2.42- and 3.58-fold respectively, and increased root length [53]. The dry weight of shoots was increased by 0.55- and 0.65-fold when it was inoculated with Cupriavidus SaCR1 and Ochrobactrum SaCRH4 [53]. To the best of our knowledge, this was the first time that K. michiganensis TS8 was applied in promoting *M. floridulus* (Lab.) growth.

Although MR2 also exhibited PGP traits, it did not significantly promote *M. floridulus* (Lab.) growth in the pot experiment. This may be explained by the fact of this strain being an endophyte, which may make its survival or growth in soil difficult. Another explanation is that MR2 could not express its PGP traits under Cd stress. TS8 was isolated from soil and exhibited better PGP traits than MR2 (Figure 1), which may make it easier to colonize in the soil and promote plant growth through producing plant-growth hormones. Previous studies found that a native bacterial consortium limited plant growth (*Agrostis capillaris* L., *Deschampsia flexuosa* (L.), *Festuca rubra* L., *Helianthus annuus* L., and *Euphorbia pithyusa*), which may be related to the establishment of bacterial communities [54].

Microbial-enhanced phytoremediation in Cd-contaminated soil is a practical on-site approach [55]. Different plant–microbe combinations are beneficial for plant growth and optimization of bioremediation of metal(loid)s [56]. It was reported that the combination of Phalaris arundinacea and microbes effectively increased metal(loids) accumulation and was beneficial for degrading polyaromatic hydrocarbons when remediating municipal sewage sludge [57]. We investigated the effects of L. jeotgali MR2 and K. michiganensis TS8 on *M. floridulus* (Lab.) ability of remediating Cd, and found that TS8 reduced the total Cd concentration in the rhizosphere soil from 10 to 5.08 mg/kg, showing an increase in remediation efficiency of about 8.1% (49.23%) if compared to the control (45.53%). In contrast, no significant effects on the Cd concentrations of plant tissues (roots, stems, and leaves) were observed. This result might be due to the high biomass of the TS8-inoculated plants. Previous research showed that PGPB with Cd tolerance could promote plant growth and increase the efficiency of phytoextraction and phytostabilization [10,58]. Therefore, the increased plant biomass might be positively related to the accumulation of metal(loid)s [59]. Previous studies also reported that PGPB might not increase the concentration of metal(loid)s in plant tissues; however, it could increase plant biomass and increase the total amount of metal accumulation in plants, thereby enhancing phytoremediation. For instance, Pseudomonas tolaasii ACC23 did not influence Cd accumulation in the root and stem of B. napus, but increased the biomass [60]. Raoultella sp. X13 promoted pak choi growth and improved production, while it decreased the Cd concentration in plant tissues [55].

However, although MR2 could not significantly decrease the Cd contents in soils compared with the CK group, it could enhance Cd translocation from root to the aboveground tissues. The Cd concentration in *M. floridulus* (Lab.) leaves inoculated with the strain MR2 was higher than in plants inoculated with the stain TS8 (Figure 6). The reason may be that MR2 can produce more acid compounds than TS8 (Figure 2), decreasing the soil pH. The effect of pH on the solubility of metal(loid)s has been fully demonstrated [61]. Low pH may increase water-soluble Cd in the soil, and it is easier to be absorbed and transported by plants. Some PGPB strains might produce low molecular weight organic acid, such as salicylic acid, citric acid, and oxalic acid, reducing the pH value in the micro-environment and playing essential roles in improving the mobility of metal(loid)s by forming a complex [62]. For instance, oxalic acid can induce the accumulation of Cd in *S. alfredii* [16]. Another reason may be that MR2 is a root endophyte, and it may easily colonize plant roots and help plants transfer Cd from roots to stems or leaves through some mechanisms, for example, stimulating the expression of plant transport protein [59]. Previous studies demonstrated that endophytes played essential roles in overcoming adverse environments [42]. They could also increase the efficiency of phytoremediation to translocate metal(loid)s from underground to aboveground tissues [63]. For instance, the inoculation of endophyte *Pseudomonas* sp. Lk9 improved the availability of metal(loid)s in the soil and increased the phytoextraction rate of Cd (17.4%), Zn (48.6%), and Cu (104.6%) by *Solanum nigrum* L. [62]. Simultaneously, Cd accumulation in the aboveground and underground parts of *B. juncea* increased by 1.33- and 7.45-fold with *Burkholdria* SaMR10, and processing for *Sphingomonas* SaMR12 increased by 1.61- and 9.93-fold, respectively [53]. Besides, other reports found that PGPB *Photobacterium* spp. strain MELD1 stimulated the *Vigna unguiculate*-phytoextraction process of multi-metals [64,65], and PGPB *Bacillus xiamenensis* PM14 stimulated the *Linum usitatissimum* L.-phytoextraction process of mercury [66]. Our results indicate that MR2 could enhance the *M. floridulus* (Lab.)-phytoextraction of Cd.

However, inoculated endophytes are not the only ones affect the metal(loid)s translocation from roots to aboveground tissues [59]. It is reported that if the bacterium, which was isolated from soil, colonized in the plant tissues after inoculated, it might also affect metal(loid)s translocation. For instance, Aeromonas sp. VITJAN13 [65], Bacillus gibsonii (PM11) [66], Lysinibacillus varians strain KUBM17 [67], and Aeromonas caviae NM04 [68] were isolated from the rhizosphere soil of plants in contaminated areas, and could all promote Cd translocation.

Overall, TS8 could enhance *M. floridulus* (Lab.) growth, and MR2 could promote Cd translocation from roots to leaves. Therefore, we suggest that *Miscanthus* spp. when co-inoculated with PGPB MR2 and TS8 may be more efficient in phytoremediation. Further studies will explore the co-inoculation of TS8 and MR2 with *M. floridulus* (Lab.) in remediating multi-metal(loid)s contaminated soil and focus on the colonization of inoculated strains in soil and plant tissues. Additionally, it is also necessary to investigate whether MR2 and TS8 could promote the expression of the related metal(loid)s-translocation proteins of plants.

## 5. Conclusions

In this study, 41 bacterial strains were isolated from 5 dominating plants in a Cd-contaminated mining area (Huayuan, Hunan, China). The strains K. *michiganensis* TS8 and *L. jeotgali* MR2 showed good PGP traits and were selected to synergize with *Miscanthus* to remediate soil Cd contamination. The results revealed that TS8 promoted *M. floridulus* (Lab.) growth and decreased the total Cd contents in soils. Although MR2 did not significantly decrease the Cd content in soils, MR2 could be considered to increase plant phytoextraction potential. PGPB were combined with energy plants to remediate Cd-contaminated soil through promoting plant growth and improving the capacity of phytoextraction, which can achieve economic and environmental benefits. Our further studies will focus on the co-inoculation of TS8 and MR2 to *M. floridulus* (Lab.) in remediating multi-metal(loid)-contaminated soil.

## Figures and Tables

**Figure 1 plants-10-00912-f001:**
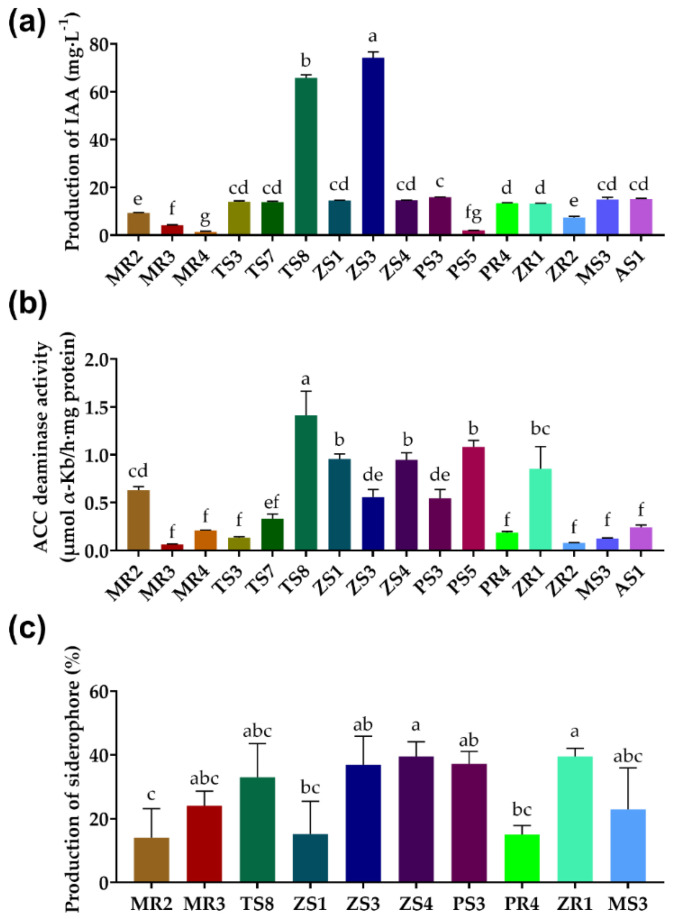
PGP traits of bacterial isolates. (**a**) IAA production, (**b**) ACC deaminase activity, (**c**) siderophore production. The data were expressed as means ± standard deviation (*n* = 3). Different letters represent significant differences between groups at the 5% level (ANOVA and Tukey’s test). IAA: indole−3−acetic acid, ACC: 1−aminocyclopropane−1−carboxylic acid.

**Figure 2 plants-10-00912-f002:**
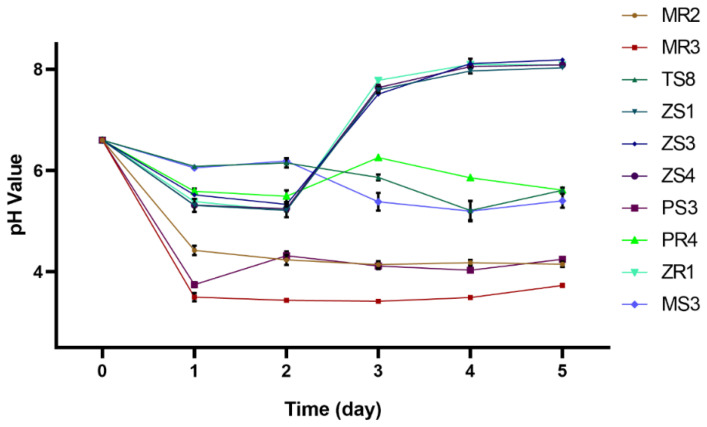
The variations of pH value after metabolizing glucose by screened bacterial isolates. The data were expressed as means ± standard deviation (*n* = 3) and the statistical analysis was calculated by ANOVA and Tukey’s test.

**Figure 3 plants-10-00912-f003:**
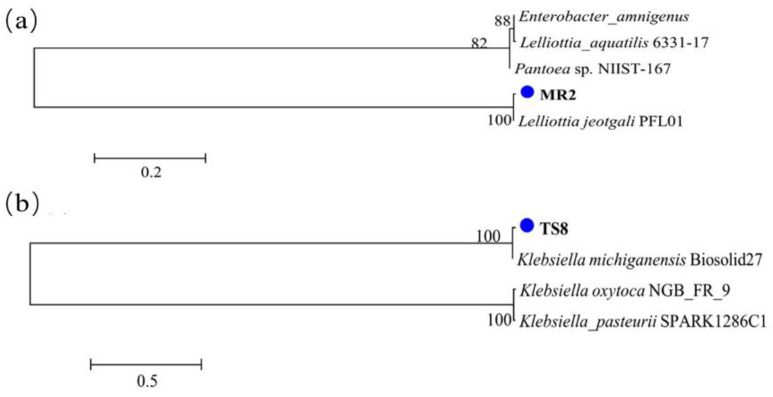
Neighbor-joining tree based on 16S rRNA gene sequences. (**a**) MR2 and (**b**) TS8.

**Figure 4 plants-10-00912-f004:**
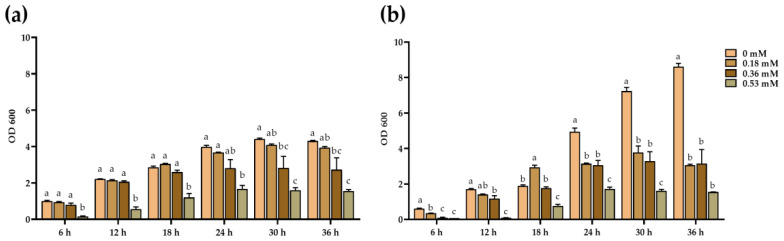
Optical density of selected bacteria at 0, 0.18, 0.36, and 0.54 mM Cd concentration during different incubation times. (**a**) MR2 and (**b**) TS8. The data were expressed as means ± standard deviation (*n* = 3). Different letters represent significant differences of absorbance between different Cd concentrations between groups at the 5% level (ANOVA and Tukey’s test).

**Figure 5 plants-10-00912-f005:**
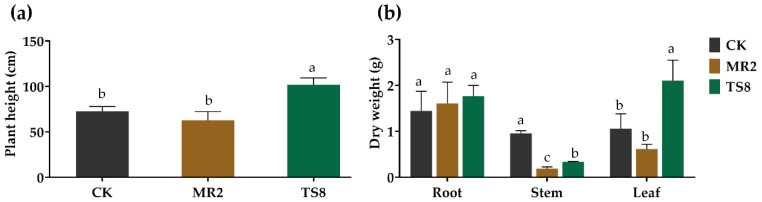
Influence of inoculation with TS8 and MR2 on *M.*
*floridulus* height (**a**) and dry weight (**b**). “CK,” without PGPB inoculation; “MR2,” Lelliottia jeotgali MR2 inoculation; “TS8,” Klebsiella michiganensis TS8 inoculation. The data were expressed as means ± standard deviation (*n* = 3). Different letters represent significant differences between groups at the 5% level (ANOVA and Tukey’s test).

**Figure 6 plants-10-00912-f006:**
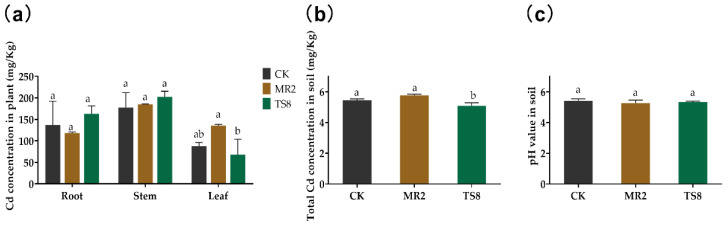
Cd concentration in plant tissues/soil and soil pH. (**a**) Effect of MR2 and TS8 inoculation on Cd uptake by *M. floridulus* root, stem, and leaf grown for two months in the contaminated soil. (**b**) Cd concentration in soil after phytoremediation. (**c**) pH value in soil after phytoremediation. The data were expressed as means ± standard deviation (*n* = 3). Different letters represent significant differences between groups at the 5% level (ANOVA and Tukey’s test).

**Table 1 plants-10-00912-t001:** N_2_ fixation, P solubilization, and K solubilization of screened PGPB. The data are expressed as means ± standard deviation (*n* = 3). Different lowercase letters represent significant differences between groups at the 5% level (ANOVA and Tukey’s test).

Strain	N_2_ Fixation	P Solubilization	K Solubilization
ZS1	1.96 ± 0.15 ^bcd^	1.52 ± 0.14 ^c^	1.97 ± 0.38 ^a^
ZS3	2.00 ± 0.34 ^abc^	2.11 ± 0.19 ^abc^	-
ZS4	1.30 ± 0.05 ^d^	2.63 ± 0.20 ^abc^	1.96 ± 0.41 ^a^
MR2	1.86 ± 0.33 ^bcd^	1.68 ± 0.02 ^bc^	2.09 ± 0.27 ^a^
MR3	2.08 ± 0.07 ^ab^	3.17 ± 1.15 ^ab^	-
MS3	1.36 ± 0.12 ^cd^	3.39 ± 0.67 ^a^	-
PR4	1.44 ± 0.05 ^bcd^	3.35 ± 0.49 ^a^	1.78 ± 0.19 ^a^
PS3	2.66 ± 0.48 ^a^	2.24 ± 0.21 ^abc^	-
ZR1	1.62 ± 0.13 ^bcd^	2.67 ± 0.67 ^abc^	1.37 ± 0.17 ^a^
TS8	-	1.67 ± 0.29 ^bc^	2.33 ± 0.58 ^a^

**Table 2 plants-10-00912-t002:** Cadmium maximum tolerance concentration of screened bacteria.

Strain	MTC (Cd mM)
ZS1	0.71
ZS3	1.07
ZS4	1.07
MR2	1.07
MR3	0.71
MS3	1.07
PR4	0.71
PS3	0.71
ZR1	1.78
TS8	0.71

**Table 3 plants-10-00912-t003:** Bioconcentration factors and translocation factors of *M. floridulus* grown in the Cd-contaminated soil inoculated with *Lelliottia jeotgali* MR2 and *Klebsiella michiganensis* TS8. Different letters represent significant differences between groups at the 5% level (ANOVA and Tukey’s test).

Treatments	BCF	TF
CK	26.89 ± 6.85 ^a^	0.82 ± 0.33 ^b^
MR2	22.14 ± 0.54 ^a^	1.25 ± 0.03 ^a^
TS8	23.60 ± 4.23 ^a^	0.54 ± 0.20 ^b^

## Data Availability

Not applicable.

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
