# Peer review of "Role of Two Plant Growth-Promoting Bacteria in Remediating Cadmium-Contaminated Soil Combined with Miscanthus floridulus (Lab.)"

_plants, 2021, doi:10.3390/plants10050912_

Round 1
Reviewer 1 Report
The manuscript “Role of Two Plant Growth-Promoting Bacteria in Remediating Cadmium-Contaminated Soil Combined with Miscanthus floridulus (Lab.)” by Liu et al. describes the characterization of bacteria isolated from the rhizosphere and endosphere of 5 plant species usually common in the Pb-Zn mining areas. Based on different characteristics, namely the ability to synthesize IAA, ACC deaminase activity and production of siderophores, among others, 2 strains, one isolated from the rhizosphere soil and another from the root endosphere, were selected to assess their potential in the phytoremediation of Miscanthus floridulus Lab. plants grown on soil artificially contaminated with Cd for 60 days.
The manuscript presents interesting results and of interest for research in this area, although it is only realized that the tests were done on soil artificially contaminated with Cd in the M&M section. This indication should be evident in the abstract. Notwithstanding the above, Abstract and Introduction sections are very well written, but the objectives at the end of the introduction section are not very clear, I think it is just a matter of English.
However, the article needs some revisions, mainly in terms of English and in the presentation of the results as well as in the discussion, which should be made from an integrative perspective of the different results and not focused on the results as being independent and isolated. In addition, there is a negligence or hurry in writing the manuscript. Various typos, lack of spaces, italics among other errors that are wrongly found throughout the text.
Minor comments
L78-88- The statement of the objectives of this study needs to be improved.
L 91-92- This sentence does makes sense. Please re-write it.
L118- liquid
L117-121 Production of IAA should be tested in minimal media and not rich medium since it may contain tryptophan. In this case, controls should be had to be used, i.e., LB without tryptophan.
L148. How was the pH measured?
L155-162- I cannot understand. Please, re-write the paragraph
L169- Please re-write the sentence
L213-215- I cannot understand why this was performed. Please, explain it.
L-219-222 These two sentences should be moved to the subsection above. This is not statistical analysis.
L226- Why “3.1- isolation” if isolation is not mentioned in this subsection
L231- α-Kb need to be spell out
Figure 4- Why an outgroup and other related type strains were not included in the phylogenetic tree
Author Response
谢谢您,对于您的评论。对评论的回复已上传。请检查出来。

Reviewer 2 Report
Title of manuscript to accept
Abstract
Correct
Key words
Please remove contamination, because Cd is contomination in the soils ....
Introduction
- 62 – I suggested add paper about high yield of Miscanthus and using plants to phytoremediation .....
For instance:
Antonkiewicz J., Kołodziej B., Bielińska E.J., Popławska A. 2019. The possibility of using sewage sludge for energy crop cultivation exemplified by reed canary grass and giant miscanthus. Soil Science Annual, 70, 1, 21-33. DOI: https://doi.org/10.2478/ssa-2019-0003
Material and methods
- 177 Please provide the available P in g or mg/kg of soil – Please see K and N
- 183 - Why was 10 mg of Cd/kg - it was contamination of soils from Hunan ?
- 201 – Too short dried plant material ?
Results
You can add information about amount Cd uptake by energy plants from 1 hectare
- 393 - Please see paper about stimulated microbial on fitoremediation
Conclusion
Correct

Author Response
Thank you for your review. The response to the comments have been uploaded. Please check them out.
Reviewer 3 Report
The manuscript takes up an interesting and important issue, however, the text requires major revision as listed here and highlighted on the attached copy. The main objection is about chosen of bacteria strains for individual analyzes, especially these ones that concern the general bacteria features. As a consequence, the experimental scheme is strongly unclear and particular parts of research seem to be unrelated to each other. It is difficult to guess why MR2 and TS8 were finally tested. In my opinion, additional control treatment without Cd application to the soil should be provided. The effect of both MR2 and TS8 inoculation could strongly improve the manuscript quality. There are also some methodological questions that should be explained before the approvement of the manuscript for publication. What is more, the discussion is slightly descriptive and do not show any mechanisms why and how bacteria influence on Miscanthus growth and phytoremediation capacity. By the way, the improvement of phytoremediation plant efficiency was not obtained since differences between particular treatment in Cd accumulation was statistically insignificant. Maybe results from this part should be express for plant biomass?
Finally, I recommend to check the text carefully, since there is many typing errors.

Author Response

(The authors gave the same response as above.)

Reviewer 4 Report
I have just some comments to improve stile as well as questions about explanation of the methods.
Line 44: "....approaches. In general, phytoremediation includes ...."
Line 50: " .... some studies found that another technology ..."
Line 52: " .... Mycorrhizal fungi helped plants to absorb nutrients and to improve ...."
Line 97: that means, the extract of 1 g of soil was diluted to 10000 litres of water ?
Line 120: please explain the contents of Sackowski´s reagent
Line 122: please explain the abbreviation "DF"
Line 130: obviously, you get ACC de-aminated (and oxidized by HO.) to alpha-ketobutyric acid, okay? Explain the reaction
Line 130: separate "unit" and "siderophore" by a new paragraph
Line 133: Why Chromazurol S? To my knowledge, siderophores are chelators for Fe, and Chromazurol S yields a colour reaction with Fe (and many others) in weak acid to neutral solution. Thus figure 1c shows the competition between the siderophore and the colour reagent - okay? Where does the Fe come from ?
Line 140: separate "100" and "The N2 fixation" by a new paragraph
Line 144: P-solubilization and K-solubilization is not clear. Which compounds get solubilized? Is this a P and K of bacterial origin, or of added soil, or else?
Line 177-179: what is the "available fraction" for you? Is it water, Ca-acetate-lactate, EDTA, Mehlich, NaHCO3, aqua regia ? 1 % of available K seems too high - please check!
Line 184: "....Plants which had been grown for 30 days ...."
A general question about the pot experiments: Miscanthus is known to incorporate much mobile silica to improve stiffness and hardness. How about potentially available silica in your soil ? How many replicates will be possible? (Tropical red soils are known to be low in silica)
Figure 1c: What is 100% of siderophore production ?
FIgure 5: "OD" means optical density ? This is not an absorbance ?
Table 4: define " translocation factor TF"
Line 373: TS8 showed the ability to solubilize P and K: did the bacteria dissolve themselves, or an added solid compound, or the test soil ?
Line 423: LMOAW maybe means "low molecular weight organic substances" - please define
Line 409: "Therefore, the increased ..."
Author Response

(The authors gave the same response as above.)

Round 2
Reviewer 1 Report
I appreciate the authors' response and the changes in the manuscript. Thank you.
Author Response
Thank you for your comments.
Reviewer 3 Report
Thank you for your detailed responses to my comments. Good luck with your future research!
Best regards,
Reviewer
Author Response
Thank you for your comments.
This manuscript is a resubmission of an earlier submission. The following is a list of the peer review reports and author responses from that submission.
Round 1
Reviewer 1 Report
The manuscript plants-1095702 has been improved; however, 2 questions still need attention.
- What do the Authors mean writing “For quality assurance and quality control, Cd content in the reference material GBW(E)083179 was certified by the National Tobacco Quality Supervision and Inspection Center in China (Table 1).” The reviewer thank that it should be “For quality assurance and quality control, Cd content in the reference material GBW(E)083179 certified by the National Tobacco Quality Supervision and Inspection Center in China was determined (Table 1).
- The reviewer think that the Authors writing “Digestion” in Table 1 mean the result of Cd determination in Their laboratory
Author Response
Thank you for your review and comments, the English of the manuscript has been modified by Editage.
Point 1: What do the Authors mean writing “For quality assurance and quality control, Cd content in the reference material GBW(E)083179 was certified by the National Tobacco Quality Supervision and Inspection Center in China (Table 1).” The reviewer thank that it should be “For quality assurance and quality control, Cd content in the reference material GBW(E)083179 certified by the National Tobacco Quality Supervision and Inspection Center in China was determined (Table 1).”
Response 1: Thanks for your correction. The sentence “For quality assurance and quality control, Cd content in the reference material GBW(E)083179 was certified by the National Tobacco Quality Supervision and Inspection Center in China (Table 1).”has been revised to “For quality assurance and quality control, Cd content in the reference material GBW(E)083179 certified by the National Tobacco Quality Supervision and Inspection Center in China was determined (Table 1).”.
Point 2: The reviewer think that the Authors writing “Digestion” in Table 1 mean the result of Cd determination in Their laboratory
Response 2: Thanks for your suggestion. The “Digestion” in Table 1 has been explain to “Digestion in our laboratory”.

Reviewer 2 Report
Thank you very much for revising the manuscript and addressing all questions.
Author Response
Thank you for your review and comments, the English of the manuscript has been modified by Editage.